# Construction and Validation of Nursing Actions to Integrate Mobile Care–Educational Technology to Assist Individual in Psychic Distress

**DOI:** 10.3390/ijerph22030419

**Published:** 2025-03-13

**Authors:** Dárcio Tadeu Mendes, Priscila de Campos Tibúrcio, Geni da Mota Cirqueira, Priscila Maria Marcheti, Sonia Regina Zerbetto, Carla Sílvia Fernandes, Maria do Perpétuo Socorro de Sousa Nóbrega

**Affiliations:** 1School of Nursing, University of São Paulo, Sao Paulo 05508-220, Brazil; dtmendes@usp.br (D.T.M.); priscila.ct@usp.br (P.d.C.T.); gnizi@usp.br (G.d.M.C.); 2Integrated Health Institute, Federal University of Mato Grosso do Sul, Campo Grande 79070-900, Brazil; priscila.marcheti@ufms.br; 3School of Nursing, Federal University of São Carlos, São Carlos 13565-905, Brazil; szerbetto@ufscar.br; 4Porto School of Nursing, 4200-072 Porto, Portugal; carlafernandes@esenf.pt

**Keywords:** nursing, mental health, primary health care, educational technology

## Abstract

Psychic suffering is typical of the human condition and involves multideterminant factors in its origin, with significant influence from affective–relational–economic issues, invariably marked by negative and positive experiences. Objective: The objective of this study is to describe the process of construction and content validation of a set of nursing actions to integrate a mobile educational technology to assist individuals in psychic distress in primary health care. Methods: This was a methodological study in four stages: scope review, qualitative research, elaboration of a set of nursing actions and content validation. It was carried out from December/2022 to December/2023, with 16 Brazilian specialists, a minimum Content Validity Index of 80% and Cronbach’s Alpha (α). Results: Six sets of actions were elaborated and evaluated: nursing actions in the initial assessment of the individual in psychic distress (99% α 0.47); nursing actions towards individuals in psychic distress with complaints associated with Depressive Disorder (93.4% α 0.84); nursing actions towards individuals in psychic distress with complaints associated with Anxiety Disorder (95.4% α 0.88); nursing actions towards individuals in psychic distress with Suicidal Ideation (96.3% α 0.71); nursing actions towards individuals in psychic distress resulting from the use of psychoactive substances (99.6% α 0.77) and; nursing actions towards individuals in psychic distress as a result of grief situations (98.6% α 0.28). Conclusions: The set of actions proved to be validated and to have acceptable reliability, thus contributing to supporting the development of educational technology. The conclusions of this research highlight the possibility for nurses to conduct nursing actions in the care of people in psychic distress, in a non-specialized context. In addition, this is a resource to improve the routine mental health care of nurses who work in primary health care.

## 1. Introduction

Psychological distress can be understood as a state of mental distress or anguish that emerges in response to experiences of adversity, internal or external conflicts, and difficulties in adapting, and is typical of the human condition [1].

It involves multiple determining factors in its origin, with significant influence from affective–relational–economic issues, invariably marked by negative and positive experiences. Individuals in psychic distress report complaints that can be considered somatic, with losses that worsen their health in general due to compromised functionality and quality of life [2].

In Brazil, despite innovative initiatives to include mental health actions in primary health care (PHC), it is observed that health care is fragile for individuals with mental health problems. A study conducted in the country indicated that, despite several innovative initiatives in the PHC of people with mental health problems, challenges persist related to the organization of work, inadequate training of professionals, lack of effective structuring of the service network, deficient infrastructure and a shortage of human resources [3].

These aspects hinder the implementation of mental health actions in PHC, strengthen the reproduction of the biomedical and asylum model, which is not person-centered, and result in unnecessary referrals to specialized services [4].

Although the PHC field in Brazil is fertile for offering nursing actions in mental health, nurses who work in this context do not feel prepared to carry out therapeutic actions and/or do not believe that they carry them out, since psychological suffering is not explicit in its presentation and is less visible than organic problems [5].

In this sense, they practice behaviors that are limited to medicalization, pathologization of demands, shallow counseling and unnecessary referrals to specialized mental health services [6].

However, for mental health nursing actions to be implemented for individuals with psychological distress in PHC, and to be successful in their development, nurses need to meet individual health needs and be resolute and equitable with the care offered in other settings, which can produce autonomy and break with the asylum logic [7].

On the other hand, providing tools that help PHC nurses conduct the care process, as well as educate themselves and others in mental health, with prospects for transformations in their performance, represents progress.

Mobile Care–Educational Technology consists of a systematic set of scientific knowledge that makes it possible to plan, execute, control and monitor the entire formal and informal educational process [8].

In the technological process, the intersection between education and care is revealed during the care–education and education–care process in nursing, with the aim of building and/or strengthening the autonomy and well-being of the individual, in a specific context of the health–disease process, mediated by the relationships of/between subjects in the course of conducting professional health practices [8].

When conceived and applied based on social reality, these aspects allow for the production of more appropriate health care, which enhances and facilitates the expansion of opportunities for empowerment and autonomy in relations that mediate care [9].

In the report of Global Strategy on Digital Health (2020–2025), the World Health Organization (WHO) estimates that more than one billion people will enjoy better health and well-being with the use of appropriate digital technologies. It highlights mobile technology as a significant vehicle for the delivery of health care and encourages the development and implementation of person-centered digital solutions [10].

Health practices supported by mobile devices within the scope of PHC assist professionals in making decisions about the care to be provided [11].

The construction and validation of nursing actions for the integration of a Mobile Caregiver Educational Technology in the care of individuals with psychological distress will provide guidelines for PHC nurses to conduct care taking into account biopsychosocial aspects, without restricting other therapeutic alternatives and intervention options.

The proposal to integrate mobile technologies and educational technology in health care can be a tool for conducting care practices, allowing real-time monitoring of interventions, stimulating autonomy and self-knowledge of the person and adapting strategies according to their needs [8].

This study aims to describe the process of constructing and validating the content of a set of nursing actions to integrate mobile educational technology for assisting individuals with mental health problems in primary health care.

## 2. Materials and Methods

### 2.1. Study Design and Sample

This methodological study, involving the development and content validation of a set of nursing actions, characterizing it as exploratory and descriptive for technological production, with a quantitative approach [8], was conducted in four stages: The 1st stage was a scope review; the 2nd stage was qualitative research; the 3rd stage was preparation of the set of nursing actions by the main researcher based on the findings of qualitative research carried out with primary health care nurses; and the 4th stage was content validation using the Delphi technique. This article was prepared following the CREDES checklist in the Equator Network steps to ensure methodological quality [12] (Appendix A).

### 2.2. Step 1—Literature Review

This literature review aims to map the main concepts involved in a given area of knowledge; examine the extent, scope and nature of research in a given area; summarize and disseminate research data; identify gaps in existing research; and provide an overview of existing evidence [13].

It was developed in the five steps recommended by JBI: identification of the guiding question; identification of relevant studies; selection of studies; mapping of information; and grouping, summary and reporting of results [13].

In order to ensure the integrity of this study and its methodological rigor, the Preferred Reporting Items for Systematic reviews and Meta-Analyses extension for Scoping Reviews (PRISMA-ScR) checklist was used for review, writing [14] and registration in the Open Science Framework (OSF). 

The mnemonic PCC was used to formulate the research question and guide data collection. The PCC strategy helps to identify the main topics: population, concept and context. In this study, P (population) refers to adults with mental health disorders over 18 years of age; C (concept) refers to nursing care; and C (context) refers to primary health care. The guiding question was defined as “What is the production of knowledge about nursing actions for adults with mental disorders in the context of primary health care?”.

The following information bases were used: Latin American and Caribbean Literature in Health Sciences (LILACS), Web of Science (WoS), American Psychological Association (APA PsycINFO), Scientific Electronic Library Online (SCIELO), Cumulative Index to Nursing and Allied Health Literature (CINAHL), AGELINE, SCOPUS, Medical Literature Analysis and Retrieval System Online (MEDLINE), Virtual Health Library-VHL and Portal of Electronic Journals in Psychology (PEPSIC).

The search strategy was developed using the controlled and uncontrolled descriptors obtained in the initial search, plus the Boolean operators “OR”, “NOT” and “AND”, and keywords found in the Health Sciences Descriptors (DeCS) and Medical Subject Headings of the US National Library of Medicine (MeSH) combined with each other, according to each database (Appendix A).

The inclusion criteria were studies available in English, Spanish and Portuguese, published between January 2011 and August 2023, with a time frame of implementation of the Psychosocial Care Network (RAPS) [15]. The exclusion criteria were review articles, reflective studies, reviews, editorials, dissertations, monographs, theses, abstracts in event proceedings, duplicates and studies that do not address the topic relevant to the objective of the review.

Duplicate articles were excluded using Endnote software 21. The searches took place in 2023. The relevance of the included studies was verified by two independent reviewers. It is noteworthy that the methodological quality of the primary studies was not assessed, since this aspect is not taken into account in literature reviews [16].

To organize and extract the data, the authors created a spreadsheet in Microsoft Excel with database/country of origin, year of publication, title, authors and nature of the study/sample. As these are public domain data, this review does not require approval from the ethics committee.

### 2.3. Step 2—Qualitative Research

The qualitative study was guided by the framework of Eric Cassell’s Theory of the Nature of Human Suffering [1], with the aim of understanding the experiences of nurses who work in primary health care caring for people with mental health problems. The study was carried out in a rural town in the state of São Paulo, Brazil, approximately 57 km from the capital, where the RAPS has 35 basic health units. The inclusion criteria were as follows: at least 1 year of experience in PHC and direct involvement in care. Exclusion criteria were as follows: nurses performing administrative activities [17].

### 2.4. Step 3—Preparation of Nursing Actions (Content)

The actions were prepared based on the findings of the literature review, qualitative research, national and international manuals and the relevant literature on possible actions for individuals in psychic distress in PHC, and the actions were developed as shown in Appendix A.

### 2.5. Step 4—Content Validation Using the Delphi Technique

The content was validated using the Delphi technique in relation to nursing actions for people with mental health problems. It consists of a systematic method that aims to reach a consensus among experts on a given topic through multiple rounds. The group of experts is composed of professionals engaged in the area in which the study is being conducted and who are motivated to think more about the subject in question, considering that they can implement the topic addressed [18].

#### 2.5.1. Period

The study operationalization phases were as follows: scope review (November and December 2022 and August and September 2023); qualitative research (January to March 2023); elaboration of the set of nursing actions (July and August 2023); content validation (September to December 2023).

#### 2.5.2. Population, Selection Criteria and Sample

To compose the panel of experts, 16 professionals from the PHC area were selected. The sample was determined following recommendations from the literature of at least 10 individuals, without the need to calculate statistical representativeness, taking into account the technical quality of the experts and the quantity of their attributes [18]. This was achieved by searching for the researchers’ resumes on a specific platform and selecting them using the snowball sampling strategy [19]. After acceptance, a link to Google Forms was sent with two sections: the 1st included access to the Free and Informed Consent Form and the 2nd included access to the questionnaire. Two rounds were carried out, with a 15-day deadline for responses (Questionnaire S1).

The specialists included obtained five or more points according to the adapted Fehring criteria [20]: doctorate in nursing (five or four points); master’s degree in nursing (three points); completion of a dissertation or thesis in mental health (one point); experience (care or academic) of at least one year with the target audience (two points); article published in an indexed journal in the area of mental health (two points); being a specialist in mental health (one point).

#### 2.5.3. Data Collection

The questionnaire was composed of 06 sets of categories and their respective items: A. nursing actions in the initial assessment of the individual in psychic distress; B. nursing actions towards individuals in psychic distress with complaints associated with Depressive Disorder; C. nursing actions towards individuals in psychic distress with complaints associated with Anxiety Disorder; D. nursing actions towards individuals in psychological distress with Suicidal Ideation; E. nursing actions towards individuals in psychological distress resulting from the use of psychoactive substances; and F. nursing actions towards individuals in psychological distress as a result of mourning situations.

A Likert-type scale was adopted to assess the relevance of the content: 1 I totally disagree; 2 I disagree; 3 I have no opinion; 4 I agree; 5 I totally agree. In order for the experts not to feel pressured or forced in relation to the negative or positive sides of the scale, a neutral midpoint of “no opinion” was provided, and an odd classification as well as spaces for comments and suggestions were also inserted [21].

#### 2.5.4. Data Analysis and Processing

Data were organized in Microsoft Excel, Office 2016, and processed in the Statistical Package for the Social Sciences software version 26. We performed a descriptive analysis with absolute and relative frequency, means and standard deviation for the sociodemographic and work characterization of the experts. To measure the proportion or percentage of agreement among the experts, the calculation of the Content Validity Index (CVI) was applied, which measures the proportion or percentage of judges who are in agreement on certain aspects of the instrument and its components [22,23].

The score given by each expert in relation to each item was added and divided by the total number of experts who participated in the rounds. A minimum agreement of 80% between the evaluators was considered to decide on the relevance and/or acceptance of the item. The internal consistency of the action categories was assessed using Cronbach’s Alpha Coefficient (α), taking into account the following parameters: values: >0.90—excellent; >0.80—good; >0.70—acceptable; >0.60—questionable; and >0.50—poor.

### 2.6. Ethical Aspects

This study was approved by the Human Research Ethics Committee, opinion number 5,788,216, with a Certificate of Presentation for Ethical Assessment (CAAE): 62834122.4.0000.5392. The procedures adopted complied with Resolution 466 of 12 December 2012 of the National Health Council.

## 3. Results

### 3.1. Literature Review

A total of 365 potentially eligible studies were retrieved (PsicINFO = 31; Lilacs = 86; Scielo = 20; PEPSIC = 04; CINAHL = 21; AGELINE = 61; Scopus = 39; MEDLINE/Pubmed = 49; Portal BVS = 31; WoS = 23), and 85 duplicate publications detected by Endnote Web were removed. A total of 280 articles were selected for the title and abstract reading stage, of which 25 were eligible. Of these, 6 articles were excluded because they were inconsistent with the study’s objectives and 19 were selected to make up the sample because they answered the guiding question and were suitable for the purpose of this research. Figure 1 shows the results using the flow diagram of the Preferred Reporting Items for Systematic Reviews and Meta-analyses (PRISMA-ScR) extension for scoping review.

From the 19 studies included, in terms of language, all publications were conducted in Portuguese (South America—Brazil) with a study design in original research of a qualitative nature. Regarding the year in which they were published, there was a dispersion of studies, where the years 2022 and 2019 had four (42%) publications each. Following closely, the years 2020, 2015 and 2012 had two (31.6%) publications each and the years 2021, 2018, 2016, 2014 and 2011 had one (26.4%) publication each.

The studies were directed at nursing actions for adults with mental health problems in primary health care. From the 19 selected studies, n = 1 was published in Web of Science^®^ [24]; n = 1 was published in Cinahl^®^ [25]; n = 15 were published in Lilacs [26,27,28,29,30,31,32,33,34,35,36,37,38,39,40]; n = 1 was published in Scopus^®^ [41]; and n = 1 was published in Pepsic^®^ [42]. Table 1 presents the variables extracted from the studies, including the study (S), database/country of origin, year, title of study, authors and nature of the study/sample.

The studies showed that nursing care for patients with psychological distress in primary care is as follows: reception (42.10%), matrix support (31.58%), medicalization (52.63%), integrative and complementary practices–auriculotherapy (5.26%), group activities (15.79%) and nursing consultation (15.79%). This makes evident the multiplicity of actions that can be carried out by the PHC nurse in relation to the individual in psychological distress.

### 3.2. Qualitative Research and Preparation of Nursing Actions (Content)

Thirty nurses participated: they were 93.3% female, aged between 40 and 49 years (36.66%), graduated between 01 and 10 years ago (66.66%), have worked in PHC for between 01 and 10 years (66.66%) and were without any specific training in mental health (93.33%). The categories found were as follows: psychological distress and its causes; needs of the individual in psychological distress; care conduct of the team; care for the person in psychological distress in the network; and instrumentation for care [17].

### 3.3. Content Validation Using the Delphi Technique

#### 3.3.1. Characterization of Experts

The sample consisted of 16 Brazilian experts in both rounds, the majority of whom were female (93.8%), with an average age of 40 years (SD = 8.44), who had post-doctorate degrees (68.8%) and experience in validation research (56.3%) and who were members of scientific societies/departments with book chapters in the thematic area (56.3%) (Table 2).

#### 3.3.2. Assessment Rounds

Two rounds were held with the same number of experts. Even with satisfactory results obtained in the first round, the second round was chosen taking into account the considerations and suggestions highlighted by the experts. Among them, suggestions for changes and additions were analyzed and considered.

The final CVI of the two rounds in all categories remained above 80% and Cronbach’s Alpha reached values between 0.7 and 0.93. Category A had more suggestions for changes, categories D and F had additions of items, and categories B, C and E underwent few changes.

In category A, the initial CVI was 99% with α 0.47; in the second round, it increased to 98.3%, with α greater than 0.83. Of the 12 items evaluated in the first round, items 1, 3, 4, 5, 6, 7, 8 and 9 received suggestions from the evaluators and were modified in terms of description and items 13, 14, 15 and 16 were added, making a final total of 16 items (Appendix A). In category B, the initial CVI was 93.4% with α 0.84; in the second round, it increased to 96.4%, with a higher α of 0.81. Of the 33 items assessed in the first round, items 4, 6, 17, 18, 20 and 29 underwent changes; there were no additions (Appendix A).

In category C, the initial CVI was 95.4 with α 0.88; in the second round, it increased to 94.9% with α 0.91. Of the 26 items evaluated in the first round, items 1, 2, 3 and 7 underwent changes, with no addition of items. In category D, the initial CVI was 96% with α 0.71; in the second round, it increased to 95.2% and α 0.89. Of the 15 items evaluated in the first round, items 1, 5, 13 and 15 were changed, with the addition of 15 items (16 to 30), making a final total of 30 (Appendix A).

In category E, the initial CVI was 99% with α 0.77; in the second round, it increased to 96.7% and α 0.70. Of the 14 items evaluated in the first round, three were modified (3, 4 and 8), with the addition of four items (15 to 18), making a final total of 18. In category F, the initial CVI was 98% with α 0. 28; in the second round, it increased to 92.0% and α 0.93. Of the 9 items evaluated in the first round, there were no changes, but 12 items were added (10 to 21), making the final total 21 items (Appendix A).

## 4. Discussion

The set of nursing actions to support mobile educational technology for PHC nurses to intervene with people in psychological distress presents evidence of satisfactory validity, since all categories presented a Cronbach’s Alpha ≥ 0.700% and CVI greater than 90% [43].

According to the reading, all content validation requires the selection of experts with a high level of knowledge and clinical experience [44]. This guidance converges with the follow-up of the present study, in which all experts have training, theoretical knowledge, clinical experience in mental health/psychiatry and studies in product construction and validation.

By triangulating the findings of the literature review, national/international manuals, and field research, it was possible to generate a density of systematized propositions (actions) based on scientific evidence regarding the needs of people with mental health problems. This set of actions has the potential to support nurses in clinical assessment, decision-making, and mental health interventions in a more autonomous manner, outside their field of specialty, that is, in primary health care.

The structure of the set of six categories of actions and th4eir respective items represents the psychic, social, cultural and biological perspective, in the individual, family and collective context, on the premise of depathologizing and demedicalizing the subjectivity of the person in psychic suffering. The structure was developed to produce comprehensive, resolutive care, in dialog with other specialties and applied within the scope of teamwork and service networks.

Supported by validated tools and interpersonal techniques that promote coping strategies, strengthening communication and psychological and pedagogical instrumentalization, the nurse is able to lead the person to behavioral, social and emotional changes of an educational and reflective nature, helping them to obtain self-knowledge and broaden the perception about oneself and one’s mental health condition.

A study showed that PHC nurses report different difficulties in caring for people with depression, such as lack of experience, lack of time, lack of training in mental health, and lack of specific knowledge about the disorder, and suggest professional training and teamwork as means of overcoming them [45].

In this sense, the category “people in psychological distress with complaints associated with Depressive Disorder” is capable of sustaining care, since small adjustments between rounds (without adding new items) maintained actions on the assessment of anxiogenic/somatic signs and symptoms, changes in appetite, sleep, psychomotricity, difficulty in decision-making, thoughts, discouragement, tiredness and suicidal behavior, as well as several possibilities of action for the person and family.

As a spectrum of suicidal behavior, Suicidal Ideation represents the initial milestone of the suicidal process, being understood as a set of thoughts of no longer existing, taking one’s own life and wishing to die [46]. Thus, the experts’ suggestion to include fifteen items (16–30) in the category “people in psychological distress with Suicidal Ideation” strengthens the scope of interventions in the assessment of risk and conduct at an appropriate level according to the complexity and contextual particularities, including the collaborative assessment of interdisciplinary teams [40], and contributes to the objectivity, clarity and relevance of this category.

From the experts’ perspective, the category “individuals in psychic distress due to the use of psychoactive substances” includes specificities for nurses to provide care from the perspective of Harm Reduction, articulated with the other devices of the Psychosocial Care Network [47].

Caring for people who use psychoactive substances represents a gap and a challenge for PHC health professionals, including nurses. For effective and humanized care, the experts indicated structured and validated instruments for detecting disorders resulting from and stratifying people with excessive alcohol consumption (AUDIT and CAGE), respectively, and ASSIST for detecting the use of alcohol, tobacco and other psychoactive substances. They also suggested the inclusion of the Brief Intervention proposal [48].

These insertions expand and enhance nursing actions for people with problems related to the consumption of psychoactive substances in all health care settings. They can overcome the lack of technical competence of professionals who are not specialized in addictions, helping to identify needs and intervene in a population with specific demands [49].

There is little scientific evidence on how professionals working in PHC navigate their practice in situations of psychological distress due to grief [50], as evidenced in the scoping review. In this sense, the category “individuals in psychic distress due to grief” gained new prominence in the second round.

The aspects of understanding grief as an individual process, recognition of the emotions experienced and redefinition of daily life without a loved one were expanded [51], in addition to the reception and support needed to strengthen the ability to cope with the intense feelings that arise during the grieving process [52].

It is worth noting that in order to maintain objectivity in this validation study, it was necessary to exclude items from the categories “person in psychic distress with Suicidal Ideation”, person in psychological distress resulting from the use of psychoactive substances” and “person in psychological distress resulting from situations of grief”.

Finally, it is important to highlight that the conceptions of Brazilian psychiatric reform by elected individuals prioritized the community as the locus of care and that the individuals brought demands for the transformation of a practice of active deconstruction of the asylum logic of psychiatric hospitals, for the conduct of actions guided and processed entirely in the territory [53] and for services to replace asylums.

Thus, within the scope of PHC, the set of proposed nursing actions ensures that nurses have mechanisms for good clinical practice in mental health, recognizing their potential to extrapolate mental health care outside of traditionally constituted specialized settings.

It also involves nurses in caring for community mental health demands, offering and monitoring less costly individual and/or collective interventions that are capable of increasing autonomy, biopsychosocial well-being and the social reintegration of people in the process and/or with mental illness.

The complexity of carrying out this study was recognized, and the evaluation of the set of nursing actions by nurses (next stage) was considered, before their clinical application, in order to verify the level of understanding of the categories/items, their applicability to the target population and the need for adaptation [54].

Implications for nursing and health

Far from restricting the conduct of nurses, their autonomy to implement other actions not included in the general proposal stands out, as it is their responsibility to choose actions based on clinical reasoning in light of the uniqueness and response to each person’s mental illness process.

Strengths and limitations

The qualitative data originated from a social reality and in a specific context. The literature review was conducted in ten databases, in only three languages, with the possibility that other nursing actions were not included. In the validation analysis process, due to the subjectivity in the judgments of the items by Brazilian experts, this was reduced with the application of the CVI.

## 5. Conclusions

With the prospect of having virtual educational technology (mobile application), a set of nursing actions was produced and validated that are capable of providing health education and directing the execution of comprehensive mental health care by PHC nurses.

The set of nursing actions certainly supports and qualifies the process of implementing specific actions, according to the emotional and psychosocial needs routinely reported by people treated by PHC and, when applied, can reduce unnecessary referral.

## Figures and Tables

**Figure 1 ijerph-22-00419-f001:**
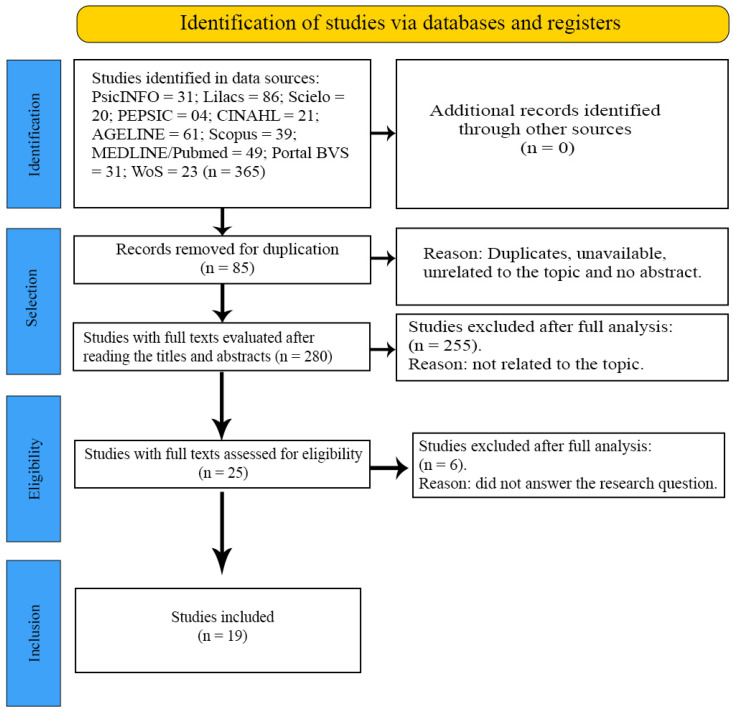
Flow diagram of the review article selection process, PRISMA-SCR. São Paulo, Brazil, 2023.

**Table 1 ijerph-22-00419-t001:** Studies included in the literature review. São Paulo, SP, Brazil, 2023.

Study (S)	Database/Country of Origin	Year	Title of Study	Authors	Nature of Study/Sample
S1 [24]	Web of Science/Brazil	2018	Mental health care technologies: primary care practices and processes	Campos DB, Bezerra IC, Jorge MSB/ Rev. Bras. Enferm	Original, descriptive and qualitative research/06 nurses
S3 [25]	Cinahl/Brazil	2022	Mental health assistance in primary care: the perspective of professionals from the family health strategy.	Cardoso LCB, Marcon SS, Rodrigues TFCS, Paiano M, Peruzzo HE, Giacon-Arruda BCC et al./Rev. Bras. Enferm	Original, descriptive and qualitative research/29 professionals from multi-professional teams
S2 [26]	Lilacs/ Brazil	2014	Professional knowledge in primary health care of the person/family in mental distress: le boterf perspective	Lucchese R, Castro P, Ba S, Rosalem V, Silva A, Andrade M et al./REEUSP	Original, descriptive and qualitative research/33 professionals from a multi-professional team
S4 [27]	Lilacs/Brazil	2021	Nurses’ contributions to matrix support for mental health in primary care.	Pinheiro GEW, Kantorski LP/Rev. Enferm. UFSM	Original, descriptive and qualitative research/15 professionals from multi-professional teams
S5 [28]	Lilacs/Brazil	2020	Primary care mental health: nurses activities in the psychosocial care network.	Nunes VV, Feitosa LGGC, Fernandes MA, Almeida CAPL, Ramos CV/REBEn	Original, descriptive and qualitative research/20 nurses
S6 [29]	Lilacs/Brazil	2019	Mental health in primary care: possibilities and weaknesses of reception.	Silva PMC, Costa NF, Barros DRRE, Silva Júnior JA, Silva JRL, Brito TS. Rev Cuidarte	Original, descriptive and qualitative research/20 nurses
S7 [30]	Lilacs/Brazil	2019	Mental health care in the context of primary care: nursing’s contributions.	Sousa SB, Costa LSP, Jorge MSB/Rev Baiana de Saúde Pública	Original, descriptive and qualitative research/07 nurses
S8 [31]	Lilacs/Brazil	2019	Mental health in Primary Care: challenges for the resoluteness of actions	Rotoli A, Silva MRS, Santos AM, Oliveira AMN, Gomes GC/Esc Anna Nery	Original, descriptive and qualitative research/30 professionals from a multi-professional team
S09 [32]	Lilacs/Brazil	2016	Nurses’ perception of care for people with borderline disorder	Cassiano APC, Silva RG, Almeida CL, Silva DA/Nursing (Sao Paulo)	Original, descriptive and qualitative research/07 nurses
S10 [33]	Lilacs/Brazil	2019	“Behind the mask of madness”: scenarios and challenges of care for people with schizophrenia in primary care.	Silva AP, Nascimento EGC, Júnior JMP, Melo JAL. Fractal/Rev. Psicol	Original, descriptive and qualitative research/10 professionals from multi-professional teams
S11 [34]	Lilacs/Brazil	2022	Perception of family health doctors and nurses on the use of auriculotherapy in Mental health problems	Silva FJB, Santos MC, Tesser CD/Interface (Botucatu)	Original, descriptive and qualitative research/44 professionals from a multi-professional team
S12 [35]	Lilacs/Brazil	2012	Mental health demands: perception of family health team nurses	Souza J, Luis MAV/Acta Paul Enferm	Original, descriptive and qualitative research/05 nurses
S13 [36]	Lilacs/Brazil	2012	Nursing care for people with mental disorders and their families in primary care	Waidman MAP, Marcon SS, Pandini A, Bessa JB, Paiano M/Acta paul Enferm	Original, descriptive and qualitative research/17 nurses
S14 [37]	Lilacs/Brazil	2020	Producing mental health care: territorial practices in the psychosocial network	Campos DB, Bezerra IC, Jorge, MSB/Trab. educ. saúde	Original, descriptive and qualitative research/60 professionals from multi-professional teams
S16 [38]	Lilacs/Brazil	2011	(Re) Building mental health scenarios in the Family Health Strategy	Oliveira FB de, Guedes HKA, Oliveira TBS de, Junior JFL./Rev.	Original, descriptive and qualitative research/13 nurses
S17 [39]	Lilacs/Brazil	2022	The role of nurses in mental health in the family health strategy	Gusmão ROM, Viana TM, Araújo DD, Vieira JPR, Junior RFS/J. Health Biol. Sci. (Online)	Original, descriptive and qualitative research/07 nurses
S18 [40]	Lilacs/Brazil	2022	Users of psychoactive substances: challenges for nursing care in the Family Health Strategy.	Militão LF, Santos LI, Cordeiro GFT, Sousa KHJF, Peres MAA, Peters AA/Esc. Anna. Nery	Original, descriptive and qualitative research/07 nurses
S15 [41]	Scopus/Brazil	2015	Psychiatric client embracement on primary health care	Silva JA, Ferreira LA, Zuffi FB., Cardoso RJ., Rezende MP, Mendonça GS/Bosci. J.	Original, descriptive and qualitative research/26 nurses
S19 [42]	Pepsic/Brazil	2015	Mental health nursing consultation in primary health care	Bolsoni EB, Heusy IPM, Silva ZF, Padilha MI, Rodrigues J/SMAD, Electronic Journal of Mental Health Alcohol Drugs	Original, descriptive and qualitative research/07 nurses

Source: Research data, 2023.

**Table 2 ijerph-22-00419-t002:** Characterization of the social and academic profile of the specialists (n = 16). São Paulo, SP, Brazil, 2023.

	N (%)	SD
Social Profile		
Gender		
Female	15 (93.8)	
Male	1 (6.3)	
Age Range		8.44
20–59 years old	15 (93.8)	
≥60 years old	1 (6.3)	
Academic Profile		
Graduation Completion		8.79
Specialization/Residency:		
Mental and Psychiatric Health	13 (81.3)	
Collective Health	2 (12.5)	
Public Health	1 (6.3)	
Master’s Degree:		
Mental and Psychiatric Health	11 (68.8)	
Collective Health	4 (25.0)	
Public Health	1 (6.3)	
Doctorate Degree:		
Mental and Psychiatric Health	10 (76.9)	
Collective Health	2 (15.4)	
Public Health	1 (7.7)	
Post-doctoral:		
No	11 (68.8)	
Yes	5 (31.3)	
Time in clinical-care as a Specialist		8.00
Time in clinical-assistance as a Master		6.67
Time in clinical-assistance as a Doctor		6.15
Time in teaching and research as a Specialist		11.63
Time in teaching and research as a Master		8.02
Time in teaching and research as a Doctor		5.86
Member of Scientific Society/Department/Chapter		
Mental and Psychiatric Health	9 (56.3)	
Collective Health	1 (6.3)	
Public Health	1 (6.3)	
Not a member of a Scientific Society/Department/Chapter in the thematic area	5 (31.3)	
Experience with construction studies and product validation		
No	7 (43.8)	
Yes	9 (56.3)	

## Data Availability

All data generated or analyzed during this study are included in this published article [and its Appendix A].

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
