# Peer review of "Construction and Validation of Nursing Actions to Integrate Mobile Care–Educational Technology to Assist Individual in Psychic Distress"

_ijerph, 2025, doi:10.3390/ijerph22030419_

Round 1

Reviewer 1 Report

Comments and Suggestions for Authors

This is a potentially important report of the development of guidance, collation of interventions available to nurses to support people suffering from psychic distress in primary care settings, and a content validation. As such, it reports an important step in the development of modern, post-asylum care community care in Brazil.

The authors designed a logical set of steps but I do not think they have reported them properly. The literature review seems like a scoping review and ought really to be reported against criteria such as https://www.equator-network.org/reporting-guidelines/prisma-scr/

The results of the review ought to form part of the results section in the paper.

I don't think this is an observational epidemiological study so the STROBE guidelines are inappropriate.

Rather, it is in the realm of a the development of a complex intervention, so other guidelines apply. For instance: https://www.equator-network.org/reporting-guidelines/tidier/

And in the UK we would be asked to consider the MRC guidelines for the development of a complex intervention - see: 

A new framework for developing and evaluating complex interventions: update of Medical Research Council guidance

BMJ 2021374 doi: https://doi.org/10.1136/bmj.n2061 (Published 30 September 2021). Cite this as: BMJ 2021;374:n2061

I don't want to suggest complicating what is in essence a concise report of what the authors have done, but I think consulting these documents will help them to reconcile and then describe the process.

The table of participant characteristics need not contain confidence limits. The authors haven't sampled from a hypothetical population of participants as an estimate of the true numbers of these categories in that population, they have just selected people based on their scores, so there is no sampling error. 95% c.i.s are better than p-values but neither is required.

Comments on the Quality of English Language

The English needs some attention in a few places where perhaps the autocorrect has failed e.g. 'It was Carry out with 30 nurses" should be 'It was carried out with 30 nurses).

Author Response

Thank you very much for your comments and suggestions. We are working on the proposed changes and would like to clarify our responses point by point, as outlined below. Please note that in the new revised manuscript, we have chosen to track changes using the “track changes” feature in Microsoft Word when making revisions in order to make it easier for editors and reviewers to find out where the changes are located. The point-by-point responses are attached to the PFD.

Reviewer 2 Report

Comments and Suggestions for Authors

Although the purpose, necessity and importance of the research is explained in the introduction, there should be a conceptual framework section in the research. There is a deficiency in this form. The concepts of Mobile Care-Educational Technology and psychic distress should be defined and described.

Research questions are missing. It is necessary to state the research questions clearly. Research questions can be included after the introduction.

The statement "In Brazil, despite innovative initiatives to include mental health actions in Primary Health Care (PHC), it is observed that health care is fragile for individuals in psychic distress." is not clear and understandable enough. The reason why it is insufficient should be explained with literature.

It would be good to state what the sampling technique is in Step 1 and Step 2. For example, in Step 2, purposive sampling seems to be based and the criterion sampling technique (having worked for at least 1 year) seems to have been used.

Table 1 looks confusing in terms of shape. The headings could be presented more systematically and the table could be more understandable.

The survey used in the study has not been explained in detail. It can be explained in a more explanatory and descriptive way, with details such as the dimensions of the survey, how it was created and by whom.

More resources need to be added, especially in the discussion section.

The article addresses an important issue and has a strong methodological structure. However, the methodology and results need to be presented more clearly and the pre-testing process needs to be detailed. The results should show the changes made, the items added and deleted in a tabular format.

Comments on the Quality of English Language

The language is generally good, but some sentences are too long and difficult to understand. It should be made clearer.

Author Response

Thank you very much for your comments and suggestions. We are working on the proposed changes and would like to clarify our responses point by point, as outlined below. Please note that in the new revised manuscript, we have chosen to track changes using the “track changes” feature in Microsoft Word when making revisions in order to make it easier for editors and reviewers to find out where the changes are located. The point-by-point responses are attached to the PFD

Round 2

Reviewer 1 Report

Comments and Suggestions for Authors

The authors have done a good job responding to my suggestions - they are correct about the appropriate reporting guidelines. All in all, the manuscript is hugely improved by the editing included in this revised version.

Reviewer 2 Report

Comments and Suggestions for Authors

The authors made all the changes I suggested. Thank you.